# Reciprocity of thermal diffusion in time-modulated systems

Jiaxin Li[1,2,10], Ying Li [1,3,4,10✉], Pei-Chao Cao[5,10], Minghong Qi[3,4,10], Xu Zheng [6], Yu-Gui Peng [7], Baowen Li [6,8], Xue-Feng Zhu [5✉], Andrea Alù [7,9✉], Hongsheng Chen [3,4] & Cheng-Wei Qiu [1✉]

The reciprocity principle governs the symmetry in transmission of electromagnetic and acoustic waves, as well as the diffusion of heat between two points in space, with important consequences for thermal management and energy harvesting. There has been significant recent interest in materials with time-modulated properties, which have been shown to efficiently break reciprocity for light, sound, and even charge diffusion. However, time modulation may not be a plausible approach to break thermal reciprocity, in contrast to the usual perception. We establish a theoretical framework to accurately describe the behavior of diffusive processes under time modulation, and prove that thermal reciprocity in dynamic materials is generally preserved by the continuity equation, unless some external bias or special material is considered. We then experimentally demonstrate reciprocal heat transfer in a time-modulated device. Our findings correct previous misconceptions regarding reciprocity breaking for thermal diffusion, revealing the generality of symmetry constraints in heat transfer, and clarifying its differences from other transport processes in what concerns the principles of reciprocity and microscopic reversibility.

[1] Department of Electrical and Computer Engineering, National University of Singapore, Singapore 117583, Singapore. [2] School of Mechatronics Engineering, Harbin Institute of Technology, 150001 Harbin, China. [3] Interdisciplinary Center for Quantum Information, State Key Laboratory of Modern Optical Instrumentation, ZJU-Hangzhou Global Scientific and Technological Innovation Center, Zhejiang University, 310027 Hangzhou, China. [4] International Joint Innovation Center, Key Laboratory of Advanced Micro/Nano Electronic Devices & Smart Systems of Zhejiang, The Electromagnetics Academy of Zhejiang University, Zhejiang University, 314400 Haining, China. [5] School of Physics and Innovation Institute, Huazhong University of Science and Technology, 430074 Wuhan, China. [6] Department of Physics, University of Colorado, Boulder, CO 80309, USA. [7] Photonics Initiative, Advanced Science Research Center, City University of New York, New York, NY 10031, USA. [8] Department of Mechanical Engineering, University of Colorado, Boulder, CO 80309, USA. [9] Physics Program, Graduate Center, City University of New York, New York, NY 10016, USA. [10]These authors contributed equally: Jiaxin Li, Ying Li, Pei-Chao Cao, Minghong Qi. ✉email: eleying@zju.edu.cn; xfzhu@hust.edu.cn; aalu@gc.cuny.edu; chengwei.qiu@nus.edu.sg

Reciprocity is a fundamental property of wave propagation[1,2] and diffusion[3], implying symmetric field transport in opposite directions. Breaking reciprocity in energy and information transport[4] is essential in components such as diodes, isolators[5–8], rotators[9], rectifiers[10,11], and circulators[12–14], spanning electromagnetics, photonics, and acoustic domains. Besides the effects of reciprocal heat transfer in static[15] and moving components[16–20], breaking the symmetry of heat transfer to achieve thermal non-reciprocity[21] is also of great importance for various applications. Devices like heat pipes and thermosyphon diodes are commonly used for thermal management and energy harvesting[22]. In addition, solid-state thermal diodes[23] and rectifiers[24] are the basic elements for thermal information processing in analogy to electronics[25,26].

In general, there are three types of approaches that break reciprocity. The first is to apply an external bias that is an odd function of time under time-reversal symmetry, like magnetic fields or mechanical motion[12,27,28]. For heat transfer, a simple external bias can be realized by introducing mass or energy fluxes that enter and leave the system with a preferred directionality. Such straightforward approach is not very practical, because it usually makes the underlying systems hardly integrable. The second is by using nonlinearity[8,29–31]. Asymmetric thermal conduction has been found in nonlinear materials[32] with temperature-dependent properties such as oxides[33] or shape-memory alloys[34,35], but the reliance on exotic materials limits its applicability and working conditions.

The third approach to break reciprocity, inspired by recent efforts in electromagnetics and acoustics[36,37], has been based on materials with time-varying properties. This scheme has received growing interest, since it is easier to be integrated and broadly applicable compared to the first two approaches. The propagation of electromagnetic waves in coupled waveguides has been shown to be non-reciprocal when the electric permittivity $\varepsilon$ is modulated with a traveling wave[36] (Fig. 1a), thanks to asymmetric mode conversions. Similar ideas have been successfully applied to thermal radiation[38] and acoustic waves[39,40]. Interestingly, time modulation can also induce asymmetric transfer of electric charge, which is essentially a diffusive process[41]. Intuitively, this is possible because the governing equation, i.e., Fick's law, contains the same Laplacian term as the wave equation. Different from wave propagation, two material parameters in the diffusion equation—the capacitance $C_e$ and electric conductivity $\sigma$ must be modulated simultaneously (Fig. 1b) to achieve this effect.

Conductive heat transfer in solids is another fundamental diffusive process, whose governing equation (Fourier's law) has the same form as Fick's law. The counterpart of electric conductivity $\sigma$ is thermal conductivity $\kappa$, while the counterpart of capacitance $C_e$ is the product of density and specific heat capacity $\rho c$. In practice, the specific heat capacity $c$ is hardly tunable, so we only consider the modulation of density $\rho$ and thermal conductivity $\kappa$ in the following discussion. It appears quite reasonable to expect thermal non-reciprocity induced by such time modulation[42], considering the continuous success of this approach in electromagnetic[36,37] and acoustic[39,40] wave propagation, and charge diffusion[41].

In this work, however, we prove theoretically and present numerical and experimental evidences that it is extremely difficult to break reciprocity in heat transfer using time modulation without resorting to external bias or special materials (Fig. 1c). This result is due to the fact that a time modulation of the density inevitably alters the governing transfer equation by taking into account the necessary mass motion **v**. Our findings indicate that diffusive heat transfer presents inherent constraints that must be carefully treated in its manipulation to break reciprocity.

## Results

**Diffusion equation under time modulation.** Heat transfer in solids is governed by the diffusion Fourier's law: $\partial(\rho cT)/\partial t = \nabla \cdot (\kappa \nabla T)$, where $T(\mathbf{r},t)$ is the temperature field, $\mathbf{r}$ is position vector, and $t$ is time. If the material is linear and not dynamic, the solutions strictly obey reciprocity[43]. In the case of dynamic materials, the density and thermal conductivity vary with time. If both parameters can be freely modulated without introducing additional effects, the Fourier's law becomes

$$\frac{\partial\big[\rho(\mathbf{r},t)cT\big]}{\partial t} = \nabla \cdot [\kappa(\mathbf{r},t)\nabla T] \qquad (1)$$

Since Eq. (1) has the same form as the time-modulated Fick's law[41], it is expected that the solution would be in general non-reciprocal[42]. However, as we will discuss in the following, it is impossible to freely modulate the density, since matter that acts as the carrier of thermal energy cannot be created or destroyed. The variation of density $\rho$ must obey the law of mass conservation, which leads to a different governing equation than Eq. (1).

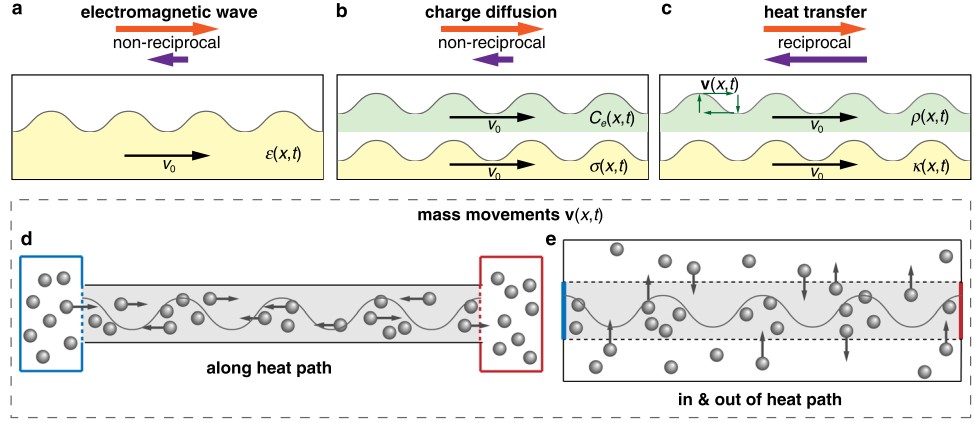

**Fig. 1 Transport processes in dynamic materials under time modulation. a** Non-reciprocal propagation of electromagnetic wave can be induced by spatial-temporally modulating the electric permittivity $\varepsilon(x,t)$ as a traveling wave (with speed $v_0$). **b** Non-reciprocal diffusion of electric charges can be induced by spatial-temporally modulating the capacitance $C_e(x,t)$ and electric conductivity $\sigma(x,t)$. **c** The reciprocity of heat transfer cannot be broken by modulating the density $\rho(x,t)$ and thermal conductivity $\kappa(x,t)$ since it is preserved by the continuity equation. Following the law, mass movements $\mathbf{v}(x,t)$ (green arrows) must exist to achieve density modulation. **d, e** Two types of mass movements (dark gray spheres): **d** along the heat transfer path (gray region). **e** moving in and out of the heat transfer path.

In continuous media, mass conservation is preserved by the continuity equation $\partial\rho/\partial t + \nabla\cdot(\rho\mathbf{v}) = 0$[44]. If the density varies with time, we inevitably expect mass movement with velocity $\mathbf{v}$ (Fig. 1c). Since thermal energy is inherent in any material, the movement introduces a convective term $\nabla\cdot(\rho c T\mathbf{v})$, which does not appear in Eq. (1). This implies that Eq. (1) can hardly be realized within a physical system without providing external energy or mass. By adding the convective term into Eq. (1), a mass-conserving diffusive heat transfer under time modulation becomes the convection-diffusion process

$$\rho(\mathbf{r},t)c\frac{\partial T}{\partial t} + \rho(\mathbf{r},t)c\mathbf{v}\cdot\nabla T = \nabla\cdot[\kappa(\mathbf{r},t)\nabla T] \qquad (2)$$

where we assume that there is no other thermal effect and that viscous dissipation is negligible. Equation (2) is a correction of Eq. (1) replacing a partial derivative by a material derivative (see Supplementary Note 1 for derivation). The detailed effects of the convective term depend on the velocity field $\mathbf{v}(\mathbf{r},t)$. In order to focus on the mechanism of time modulation, it is reasonable to only study systems at time-harmonic steady state without externally applied directional mass or energy flux. To be specific, the boundary conditions are constant, and the modulation of material parameters is periodic with time to ensure a stable (time-harmonic) field. No external mass flux exists: $\rho\mathbf{v}\cdot\mathbf{n} = 0$, where $\mathbf{n}$ is the unit normal vector at the system boundary. In time-modulated systems, it is suitable to require instead that the average external mass flux in a time period $t_0$ vanishes

$$\langle\rho\mathbf{v}\cdot\mathbf{n}\rangle = \int_0^{t_0}\rho(\mathbf{r},t)\mathbf{v}(\mathbf{r},t)\cdot\mathbf{n}\,dt/t_0 \qquad (3)$$

No accumulated external bias is a central assumption that will be used throughout our analysis.

There are only two types of setups that support density modulation without external bias: in the first case, the density is modulated by mass motion along the heat transfer path (Fig. 1d). The spheres illustrated do not represent microscopic particles but macroscopic components. There can be an exchange of mass between the system and two thermal reservoirs at both ends, but the three parts together restore the original state after a period, hence there is no net directional mass flow. In the second one, the density is modulated by mass entering or leaving the heat transfer path cyclically (Fig. 1e).

In order to determine whether a two-port system is reciprocal in diffusion, we define the thermal reciprocity based on the equivalence between steady-state global non-reciprocity and thermal diode effect[21]. Specifically, if the system is reciprocal, the heat transfer is symmetric before and after swapping the boundary conditions at two ports, then it is required that the time-averaged heat flux $\langle q\rangle$ at harmonic steady state satisfies

$$\begin{pmatrix}\langle q_{\mathrm{f},2}\rangle \\ \langle q_{\mathrm{f},1}\rangle\end{pmatrix} = -\begin{pmatrix}\langle q_{\mathrm{b},1}\rangle \\ \langle q_{\mathrm{b},2}\rangle\end{pmatrix} \qquad (4)$$

where the subscripted b and f represent the cases before and after the exchange of boundary conditions, and 1 and 2 represent the position of two ports. In the following, we prove that the heat transfer in both setups is inherently reciprocal and experimentally build a setup of the second type to validate our prediction.

**Density modulations of the first type**. The first type of modulation scheme follows the one-dimensional (1D) model shown in Fig. 2a. The density $\rho(\chi)$ and thermal conductivity $\kappa(\chi)$ of the material are $d$-periodic functions of $\chi = x - v_0 t$, so their profiles move at constant speed $v_0$ along $x$. According to the 1D continuity equation $\partial\rho/\partial t + \partial(\rho v)/\partial x = 0$, the mass flux $\rho v$ satisfies $\rho v$

$= (\rho - \rho_0)v_0 + C$, where $\rho_0$ is the average density and $C$ is a constant (See Supplementary Note 2 for derivation). According to Eq. (3), there is no accumulated mass flux through the system in a time period $t_0 = d/v_0$ (Fig. 2b). Thus, we have $C = 0$ (see Supplementary Fig. 1 for the effects of a nonzero $C$) and can solve for the velocity field as $v(\chi) = [\rho(\chi) - \rho_0]v_0/\rho(\chi)$. The 1D heat transfer then obeys

$$\rho(\chi)c\frac{\partial T}{\partial t} + [\rho(\chi) - \rho_0]cv_0\frac{\partial T}{\partial x} = \frac{\partial}{\partial x}\left[\kappa(\chi)\frac{\partial T}{\partial x}\right] \qquad (5)$$

We apply fixed temperature boundary conditions $T(0,t) = T_{\mathrm{cold}}$ and $T(L,t) = T_{\mathrm{hot}}$ (backward) or $T(0,t) = T_{\mathrm{hot}}$ and $T(L,t) = T_{\mathrm{cold}}$ (forward) at the two ends, respectively. For Eq. (5), such a symmetry can be proved by comparing the forward and backward heat fluxes. Assuming that $T_{\mathrm{b}}(x,t)$ is the solution for the backward case, while $T_{\mathrm{f}}(x,t)$ is the solution for the forward case. Given any initial conditions, both solutions at time-harmonic steady state should be unique. Their summation $T_{\mathrm{s}}(x,t) = T_{\mathrm{f}}(x,t) + T_{\mathrm{b}}(x,t)$ also satisfies Eq. (5) with boundary conditions $T_{\mathrm{s}}(0,t) = T_{\mathrm{s}}(L,t) = T_{\mathrm{hot}} + T_{\mathrm{cold}}$. It is easy to check that $T_{\mathrm{s}}(x,t) = T_{\mathrm{hot}} + T_{\mathrm{cold}}$ is a solution, and must be the unique solution thanks to the uniqueness of $T_{\mathrm{f}}(x,t)$ and $T_{\mathrm{b}}(x,t)$. The heat flux $q(x,t)$ is the sum of conductive and convective heat flux: $q(x,t) = -\kappa\partial T/\partial x + \rho c v(T - T_{\mathrm{ref}})$, where the reference temperature $T_{\mathrm{ref}}$ is an arbitrary constant that can be selected based on convention (We set $T_{\mathrm{ref}}$ equal to the lowest temperature $T_{\mathrm{cold}}$). Then the forward and backward heat fluxes $q_{\mathrm{f}}(x,t)$ and $q_{\mathrm{b}}(x,t)$ then satisfy

$$q_{\mathrm{f}}(x,t) + q_{\mathrm{b}}(x,t) = -\kappa\frac{\partial T_{\mathrm{s}}}{\partial x} + (\rho - \rho_0)v_0 c(T_{\mathrm{s}} - 2T_{\mathrm{ref}}) \qquad (6)$$
$$= (\rho - \rho_0)v_0 c(T_{\mathrm{hot}} - T_{\mathrm{cold}})$$

Averaging over time gives $\langle q_{\mathrm{f}}(x)\rangle + \langle q_{\mathrm{b}}(x)\rangle = 0$. Considering that the average heat fluxes in and out of the system should balance, we have $\langle q_{\mathrm{f}}(0)\rangle = \langle q_{\mathrm{f}}(L)\rangle = -\langle q_{\mathrm{b}}(0)\rangle = -\langle q_{\mathrm{b}}(L)\rangle$, which meets the condition for a symmetric heat transfer as in Eq. (4), and indicates thermal reciprocity.

We can also analytically solve Eq. (5). After a variable change $(x,t)$ to $(\chi = x - v_0 t, \tau = t)$, it is easy to see that Eq. (5) is periodic on $\chi$, so Floquet–Bloch theorem applies and gives (see Supplementary Note 2 for details):

$$T(x,t) = e^{iKx}f(\chi) \qquad (7)$$

where $f(\chi)$ is a periodic function with periodicity $d$, and $K$ is the Bloch wavenumber. The time-harmonic temperature solution should be a linear combination of Eq. (7). The dissipative nature of heat transfer indicates that $K$ must have nonzero imaginary parts[45]. Substituting Eq. (7) into (5), the temperature field can be analytically solved using the Fourier series of $f(\chi)$, based on the periodicity of $\rho(\chi)$ and $\kappa(\chi)$.

To verify the solution, we build a 1D model with $d = 1$ cm and $L = 10d$. The density and thermal conductivity are set to be $\rho(\chi) = \rho_0[1 + \Delta_\rho\cos(\beta\chi)]$ and $\kappa(\chi) = \kappa_0[1 + \Delta_\kappa\cos(\beta\chi)]$, where $\beta = 2\pi/d$, $\rho_0 = 2000$ kg m$^{-3}$, $\Delta_\rho = 0.3$, $\kappa_0 = 100$ W m$^{-1}$ K$^{-1}$, and $\Delta_\kappa = 0.9$. The specific heat capacity is $c = 1000$ J kg$^{-1}$ K$^{-1}$. We choose two modulation speeds $v_0 = \mu\kappa_0/\rho_0 c$ with $\mu = 1/d$ and $4/d$. Constant temperatures are set as $T_{\mathrm{cold}} = 273$ K and $T_{\mathrm{hot}} = 323$ K to generate a temperature difference $\Delta T = T_{\mathrm{hot}} - T_{\mathrm{cold}} = 50$ K. Our analytical results (solid lines in Fig. 2c, d) are well validated by finite-element simulations (scatter points in Fig. 2c, d) with COMSOL Multiphysics®. The corresponding heat flux distributions are summarized in Supplementary Fig. 2a, b, demonstrating the thermal reciprocity.

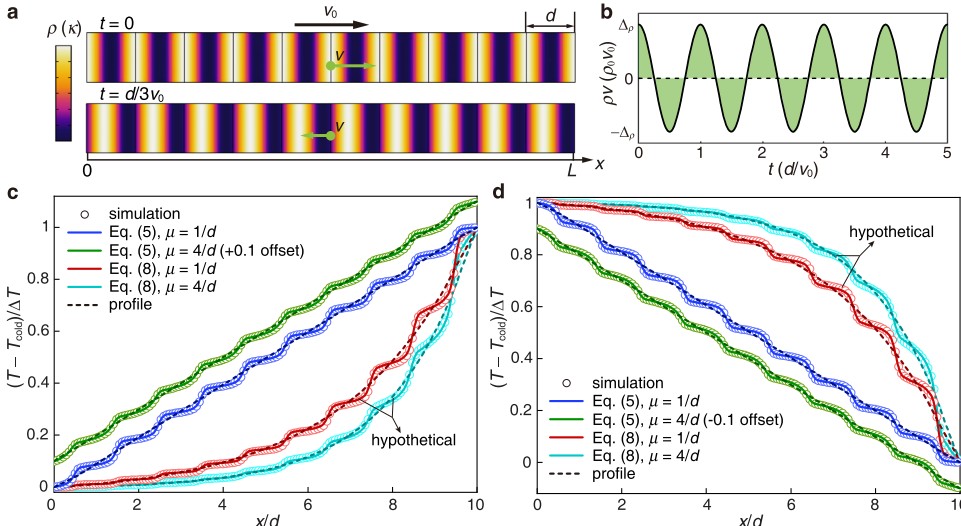

**Fig. 2 Heat transfer under 1D density modulation. a** Density $\rho$ and thermal conductivity $\kappa$ (color maps) move as traveling waves at speed $v_0$ with wavelength $d$. To achieve density modulation, actual mass movements (green arrows) at speed $v$ must exist. **b** The total mass flux in a time period should be zero to keep a cyclic and close setup. **c, d** Backward (**c**) and forward (**d**) temperature distributions of the system (Eq. (5)) at $t = Nd/v_0$ ($N$ is a large enough integer to achieve time-harmonic steady state), compared with those of a virtual system without mass movements (Eq. (8)). Scatter points are simulated results, lines are analytical results, and dashed lines are analytical solutions of the homogenized profiles. For clarity, the results of Eq. (5) at modulating speed $\mu = 4/d$ are shifted to have $\pm 0.1$ offsets.

For comparison, we also plot analytical and numerical solutions to the diffusion equation[42]

$$\rho(\chi)c\frac{\partial T}{\partial t} = \frac{\partial}{\partial x}\left[\kappa(\chi)\frac{\partial T}{\partial x}\right] \quad (8)$$

The solutions to Eq. (5) for the heat transfer are symmetric in the backward (Fig. 2c) and forward (Fig. 2d) directions for all modulation parameters characterized by $\mu$, $\Delta_\kappa$, and $\Delta_\rho$. This is in contrast with the solutions to Eq. (8) with concave/convex profiles (see Supplementary Note 3 for accurate solutions and Supplementary Fig. 2c, d for heat flux distributions). The insightful work[42] shows the possibility to generate thermal non-reciprocity at a mathematical level, with the material parameters in Eq. (8) set as space- and time-dependent functions. However, Eq. (8) is a hypothetical one that requires an external energy source, because density modulation $\rho(\chi)$ cannot be achieved at no cost, and the implementation of modulation will make the governing equation deviate from the previous theoretical design. In a virtual system following Eq. (8), there should be a difference between the average heat flux entering and leaving the system at the two ends, showing that additional energy input or extraction is required to compensate it.

**Density modulations of the second type**. Another way to achieve density modulation without net directional flow is to add/remove matter periodically through the heat transfer path. The simplest setup of such type is the two-dimensional (2D) model shown in Fig. 3a, where the heat transfer path under consideration is the transverse narrow region at $y = 0$. We assume constant temperatures on the left and right sides at $x = 0$ and $L$, while periodic boundaries are assumed on the upper and lower sides at $y = \pm d_y/2$. $\rho(x,y = 0,t)$ and $\kappa(x,y = 0,t)$ are $d$-periodic functions of $x - v_0t$, so we consider 2D distributions that are $d$-periodic functions of $\zeta = x + \eta y - v_0t$ with $\eta = d/d_y$.

Next, we consider mass motion along $y$ with speed $v_y(x,y,t)$ to locally modulate the density. According to the continuity equation $\partial\rho(\zeta)/\partial t + \partial[\rho(\zeta)v_y]/\partial y = 0$, we find $\rho v_y = (\rho - \rho_0)v_{0y} + C$, where $v_{0y} = v_0/\eta$. The derivation can be obtained in the same way as in 1D case. When $C = 0$, the case is realizable with mass oscillations

in $y$ direction, which is almost the same as the 1D case. Here, we apply periodic condition to the upper and lower boundaries and set $C = \rho_0 v_{0y}$, so that $v_y(\xi,y,t) = v_{0y}$ (Fig. 3a). It is noted that Eq. (3) is still satisfied, and this case can be realized on the side surface of a rotating cylinder. The 2D heat transfer follows

$$\rho(\zeta)c\frac{\partial T}{\partial t} + \rho(\zeta)cv_{0y}\frac{\partial T}{\partial y} = \frac{\partial}{\partial x}\left[\kappa^x(\zeta)\frac{\partial T}{\partial x}\right] \quad (9)$$

in which we consider the general case with a thermal conductivity modeled as an anisotropic tensor. Its $xx$ and $yy$ components are $\kappa^x$ and $\kappa^y$, while the off-diagonal components are assumed to be zero. Similar to the 1D case, we can prove the thermal reciprocity by analyzing the time-averaged heat fluxes along $x$ direction in forward and backward regimes, which also gives $\langle q_f \rangle + \langle q_b \rangle = 0$ and satisfy Eq. (4). The solution to Eq. (9) can be solved with a similar method as in 1D case (see Supplementary Note 4).

A practical 3D setup that realizes the time modulation of material parameters is shown in Fig. 3b, which consists of fixed and moving fan-shaped solid plates with density $\rho_A = 8390 \text{ kg m}^{-3}$, heat capacity $c_A = 375 \text{ J kg}^{-1} \text{ K}^{-1}$, and thermal conductivity $\kappa_A = 123 \text{ W m}^{-1} \text{ K}^{-1}$. Each plate spans $\pi/2$ with inner and exterior radius $R_1 = 1 \text{ cm}$ and $R_2 = 2 \text{ cm}$, and thickness $\delta = 0.25 \text{ cm} = d/16$. The total length of the system $L = 5d = 20 \text{ cm}$. Temperature boundary conditions are $T(0,t) = T_{cold}$ and $T(L,t) = T_{hot}$ (backward) or $T(0,t) = T_{hot}$ and $T(L,t) = T_{cold}$ (forward), with constant temperatures set as $T_{cold} = 273 \text{ K}$ and $T_{hot} = 323 \text{ K}$. All other boundaries are thermally insulated. Naturally, we can regard the heat transfer path as the portion of the system contained in the region $([R_1,R_2], [\pi/4,3\pi/4], [0,L])$ of the cylindrical coordinate system $(r, \theta, x)$, through which most of the heat flux is conducted. Along the $x$ direction, each moving plate is $\pi/4$ ahead of the previous moving one. All of them rotate at angular speed $-\Omega = -0.06\pi \text{ rad s}^{-1}$. As the moving plates enter or leave the heat transfer path at $y = 0$, the density $\rho(x,y = 0,t)$ and thermal conductivity $\kappa(x,y = 0,t)$ are effectively modulated (Supplementary Note 5).

We perform numerical simulations on the 2D and 3D models in Fig. 3a, b. As expected, reciprocal heat transfer is confirmed for

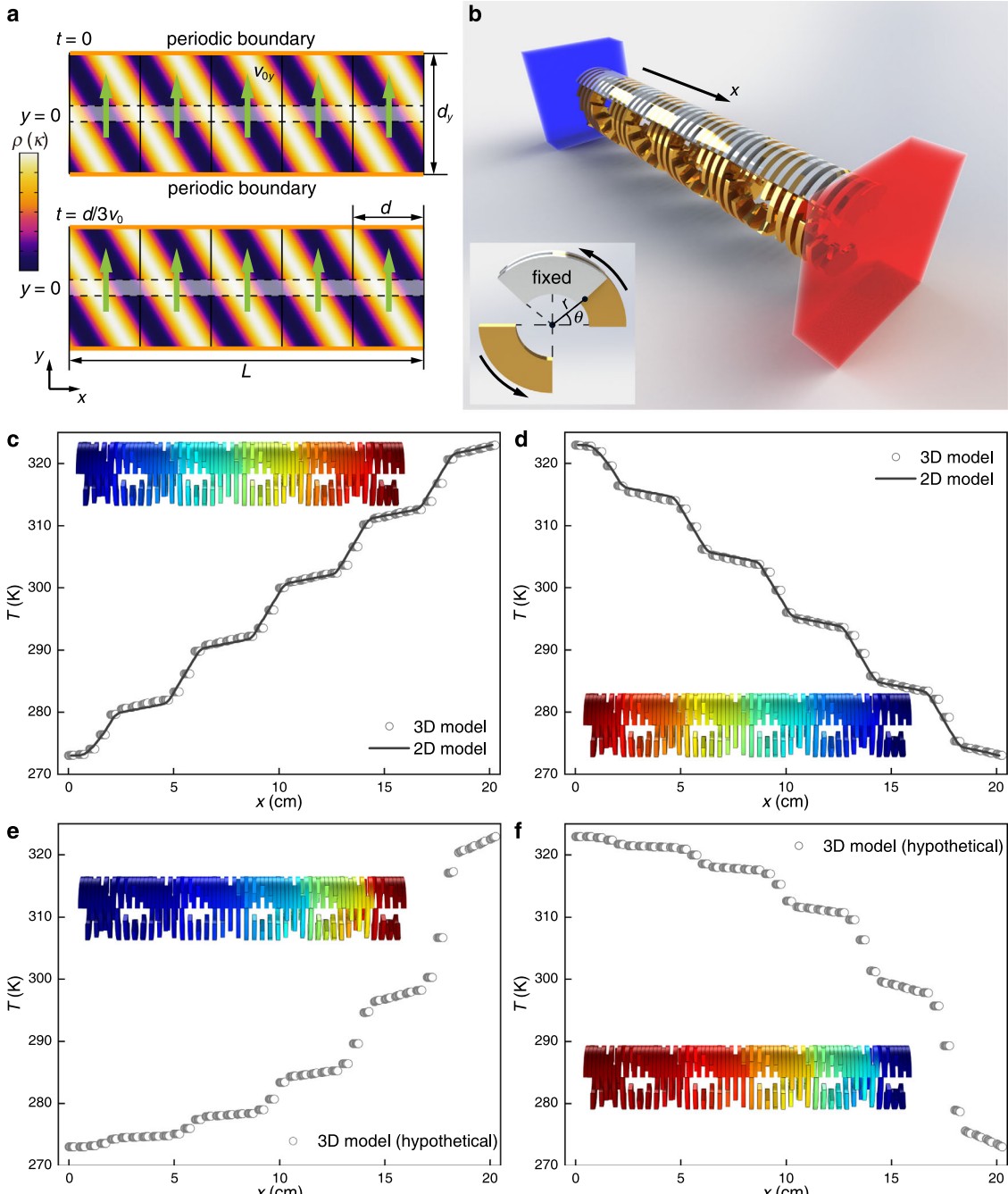

**Fig. 3 Heat transfer under 2D and 3D density modulation. a** Density $\rho$ and thermal conductivity $\kappa$ profiles (color maps) of 2D traveling waves such that the profiles move at speed $v_0$ in $x$ direction (marked by dashed lines). The density is modulated by mass movement in $y$ direction at speed $v_{0y}$ (green arrows). The upper and lower boundaries are periodic. **b** A three-dimensional (3D) model in a $(r,\theta,x)$ cylindrical coordinate system. Similar $\rho$ and $\kappa$ modulations are achieved with fixed (white) and moving (yellow) fan-shaped plates. **c-f** Backward (**c**, **e**) and forward (**d**, **f**) temperature distributions on the top line $r = R_2$, $\theta = \pi/2$ extracted from simulated results for the 2D (lines) and 3D (scatters) models at $t = N2\pi/\Omega$ ($N$ is a large enough integer to achieve time-harmonic steady state). The insets show the entire temperature distributions. The results in **e** and **f** use a hypothetical 3D model where the moving plates have time-varying masses, violating the law of mass conservation.

all considered temperature distributions along the line at ($r = R_2$, $\theta = \pi/2$) (Fig. 3c, d). The temperature distributions on the plate surfaces are also plotted in the insets (see Supplementary Movie 1 for the evolution with time and Supplementary Fig. 3a for the heat flow in and out of the system). Reciprocity can be easily shown from the symmetric temperature profiles and the identical time-averaged heat fluxes in forward and backward directions. The reason for the absence of non-reciprocity resides in the fact that in these diffusive systems the density variations are averaged out over time due to the conservation of mass and hence there is effectively no density modulation. To further illustrate this point, we performed simulations on a hypothetical model where the density of the moving plates is artificially modulated as $\rho_A(t) = \rho_A\{1 + \cos[2\Omega t - (n - 1)\pi/4]\}/2$, for the $n$-th layer of moving plates. The temperature distributions in Fig. 3e, f show concave and convex profiles, indicating non-reciprocity (see Supplementary Movie 2 for the evolution and Supplementary Fig. 3b for the heat flow).

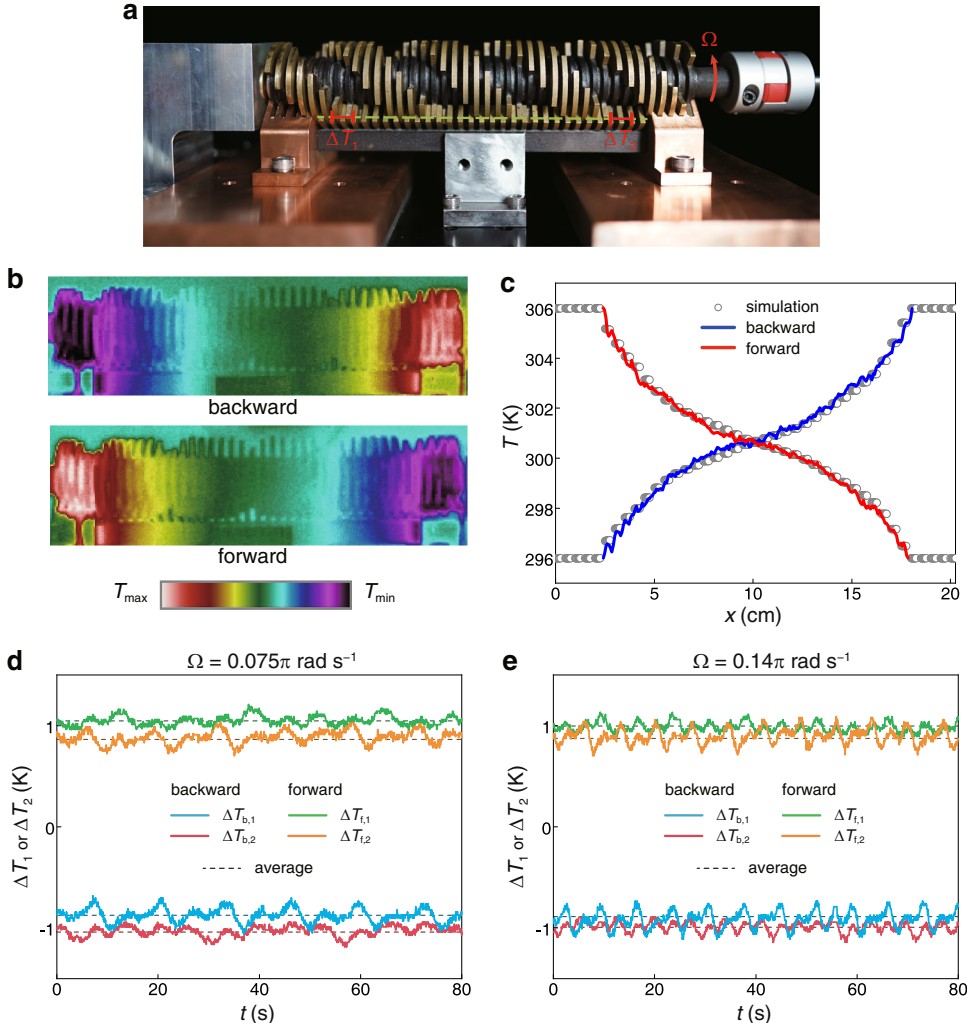

**Fig. 4 Experimental verification of reciprocal heat transfer under time modulation. a** Photo of the system built according to the 3D modulation model in Fig. 3. The movable brass plates are rotated to generate a right-ward moving profile. **b** Temperature profiles for backward and forward heat transfer. **c** Experimentally measured temperature profiles along the heat transfer path of the system (dashed green line in **a**). The scatter points refer to simulated results considering natural air convection and interface thermal resistance. **d**, **e** Experimentally measured temperature difference (between two ends of the red segment line in **a**) changing with time $t$. The angular velocity of the shaft rotated by the motor is $0.075\pi$ rad s$^{-1}$ (**d**) and $0.14\pi$ rad s$^{-1}$ (**e**). The symmetry of time-averaged temperature difference implies the reciprocity of heat flow in Eq. (4).

**Experimental verification of reciprocal heat transfer in a time-varying material.** We have implemented the 3D geometry in Fig. 3b, as shown in Fig. 4a. The system is built with fan-shape plates attached to a fixed beam and a shaft revolved by a low-speed motor (angular velocity $\Omega = 0.075\pi$ rad s$^{-1}$) to generate a right-handed rotating profile. The plates are made of brass with the same material properties and shape as in the 3D simulations. The supporting beam and shaft are made of nylon with thermal conductivity of 0.3 W m$^{-1}$ K$^{-1}$. Two copper blocks are mounted at the ends of the supporting beam to contact the heat sources directly at 296 and 306 K, generating a temperature difference $\Delta T = T_{hot} - T_{cold} = 10$ K. The blocks are slotted at the top to match the upper four rotating fan-shape plates. Different from the numerical simulations, a 0.2 mm gap is made between adjacent plates, which is filled with thermal grease (thermal conductivity 4.38 W m$^{-1}$ K$^{-1}$) for conduction and silicone oil (thermal conductivity 0.16 W m$^{-1}$ K$^{-1}$) for lubrication. Therefore, the interface thermal resistance is nonnegligible.

In addition, the natural convective heat exchange with the ambient air introduces another term $h(T_\infty - T)$, where $h$ is the heat exchange rate and $T_\infty$ is the room temperature. In order to

reduce the influence of ambient air convection, we set the system in a vacuum chamber (see Supplementary Fig. 4). Molecular pump (LF-110) and mechanical pump (BSV-16) are employed to reduce the density of air to as low as $10^{-3}$ Pa. Temperature profiles are measured through the inspection window with an infrared camera (Fotric 347). Due to the interface thermal resistance as well as natural convective heat exchange, the temperature gradients at the center are smaller compared to the linear profile (Fig. 4b). However, the two factors do not have any directionality, so the temperature profiles for the backward and forward cases are still symmetric, demonstrating reciprocal heat transfer in the time-modulated system. See Supplementary Movie 3 for videos of the rotating device and the temperature profiles. Reciprocity is further confirmed by the temperature distribution along the heat transfer path of the system (dashed green line in Fig. 4a), where the fixed plates are placed (Fig. 4c). In Fig. 4c, to compare with the experimental results, numerical simulations were performed on the 3D model with interface thermal resistance (0.2 mm thickness and 1 W m$^{-1}$ K$^{-1}$ thermal conductivity) and natural air convection ($h = 3$ W m$^{-2}$ K$^{-1}$, and $T_\infty$ was set as the ambient temperature 300.5 K.) The

experimentally measured curves for backward and forward heat transfer are not only symmetric, but also in good agreement with the simulated results. Since the heat flux in $x$ direction is proportional to temperature gradient $q_x = -\kappa \partial T/\partial x$, and the equivalent thermal conductivity is homogeneous throughout the system, the reciprocity of heat flow can be reflected from the symmetry of the temperature difference over two intervals of equal length (marked in Fig. 4a). We give the measured $\Delta T_1$ and $\Delta T_2$ in Fig. 4d, and averaging the temperature difference over time gives: $\langle \Delta T_{b,1} \rangle = -0.874$ K, $\langle \Delta T_{b,2} \rangle = -1.038$ K (backward) and $\langle \Delta T_{f,1} \rangle = 1.046$ K, $\langle \Delta T_{f,2} \rangle = 0.863$ K (forward), showing a symmetric relationship: $\langle \Delta T_{b,1} \rangle \approx -\langle \Delta T_{f,2} \rangle$ and $\langle \Delta T_{b,2} \rangle \approx -\langle \Delta T_{f,1} \rangle$. This symmetry is maintained for all angular velocities of the shaft rotated by the motor. As another example in Fig. 4e, the angular velocity $\Omega$ is changed to $0.14\pi$ rad s$^{-1}$ and the time-averaged temperature difference becomes: $\langle \Delta T_{b,1} \rangle = -0.887$ K, $\langle \Delta T_{b,2} \rangle = -0.992$ K (backward) and $\langle \Delta T_{f,1} \rangle = 0.996$ K, $\langle \Delta T_{f,2} \rangle = 0.872$ K (forward). In order to further demonstrate the reciprocity, we give the numerical results of the heat flow $Q$ in and out of the system (see Supplementary Fig. 5). It is easy to check that $\langle Q_{b,1} \rangle = -\langle Q_{f,2} \rangle$ and $\langle Q_{b,2} \rangle = -\langle Q_{f,1} \rangle$, satisfying the condition for thermal reciprocity in Eq. (4). This reciprocal result exhibited in the experiment is a representative of the general situation where the modulation parameters (e.g., rotating angular velocity) are changed arbitrarily.

## Discussion

In this paper, we have shown that physical systems preserve thermal reciprocity under time modulation as a result of a convection correction introduced by density modulation. For other processes governed by the momentum equation (e.g., wave propagation) or the continuity equation (e.g., charge diffusion), the driving approach of time modulation does not alter the form of the governing equation. For heat transport, however, it is a well-known fact that a material derivative must be used if there is a transport of mass. As such, a convection correction should be added in the governing equation, resulting in the reciprocity in heat transfer. Though only several simulation and experiment examples are given in this work, the conclusion of reciprocity is theoretically rigorous. One evidence is that the proven symmetric relationship between the forward and backward heat flux $\langle q_f \rangle = -\langle q_b \rangle$ always exists, regardless of how the material parameters are modulated. Even if the modulated material parameters are asymmetric in spatial distribution, the system is reciprocal in heat transfer (Supplementary Fig. 6).

While we did not explicitly discuss the case of a varying specific heat capacity $c$, it is easy to recognize that our arguments equally apply to this scenario. First of all, if the variation is simply a consequence of mechanical motions, the material derivative of $\rho c$ must be zero, which gives $\partial(\rho c)/\partial t + \nabla \cdot (\rho c \mathbf{v}) = 0$. Therefore, Eq. (2) and the following analysis still apply. Second, if the specific heat capacity of the material can really be modulated at will, it is possible to avoid the convective term and thereby in principle achieve non-reciprocity. However, this is only possible in very limited scenarios, such as using caloric materials that undergo phase transitions in the presence of electric[46] or magnetic[47] fields. In addition, there are schemes that can break thermal reciprocity beyond our assumptions. Though thermal reciprocity is preserved as the boundary conditions are maintained at two constant temperatures, the non-reciprocity of thermal wave[48,49] under space-time modulation may occur when the boundaries are set as periodically oscillating heat sources[50]. Another trivial way to generate non-reciprocity is introducing directional mass/energy bias while modulating the parameters.

Our work suggests that thermal reciprocity has more fundamental resilience than other transport mechanisms. These findings may have important implications for the design of thermal devices and other dissipative wave propagation systems.

## Data availability

All technical details for producing the figures are enclosed in the Supplementary Information. The experimental data generated in this study are provided in the Source Data file. Additional data that support the findings of this study are available from the corresponding author (C.-W.Q.) on request. Source data are provided with this paper.

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

## Acknowledgements

C.-W.Q. acknowledges the support from the Ministry of Education, Singapore (Grant No. R-263-000-E19-114). Y.L. acknowledges the financial support of the National Natural Science Foundation of China (Grant No. 92163123). X.-F.Z. and P.-C.C. acknowledge the financial support of the National Natural Science Foundation of China (Grant Nos. 11690030, 11690032, and 11674119), and the Bird Nest Plan of HUST. A.A. acknowledges the support of the Air Force Office of Scientific Research and the Department of Defense. H.C. acknowledges the support from the National Natural Science Foundation of China (Grant Nos. 61625502, 61975176, and 11961141010).

## Author contributions

J.L., Y.L., and C.-W.Q. conceived the idea. J.L. proposed the analytical model. J.L. and Y.L. performed the theoretical derivations. J.L., Y.L., Y.-G.P., and H.C. designed and performed the numerical simulations. J.L., Y.L., P.-C.C., and M.Q. designed and performed the experiments. X.Z., B.L., and A.A. contributed to the analysis on heat flux. J.L. and Y.L. wrote the manuscript. Y.L., X.-F.Z., A.A., and C.-W.Q. supervised the work. All the authors discussed and contributed to the manuscript writing.

## Competing interests

The authors declare no competing interests.
