## [Peer Review File · Nature Communications]

Reviewers' Comments:

Reviewer #1:

Remarks to the Author:

Authors present a work on "Reciprocity of thermal diffusion in time-modulated systems". From the abstract, they badly formulate the motivation by saying "Quite surprisingly, here we show that, from a practical point of view, time modulation cannot generally be used to break reciprocity for thermal diffusion." Practicability does not contradict the possibility or not!

Define the reciprocity mathematically in diffusion. It is not so clear compare to EM or acoustics. There is no wavevector and thus you need more to define the problem you solve and in particular the BC (even the orthogonal once to the direction of heat flux). What about the parity of the system?

I find the paper interesting but very badly presented and I get almost not convinced by the experimental part that is not well explained. Must be reformulated.

Reviewer #2:

Remarks to the Author:

This work suggests that thermal reciprocity has more fundamental resilience than other transport mechanisms. The findings will be important for the thermal system design and other dissipative wave propagation systems. I suggest to publish this paper in Nature Communication after minor revision with following comments

- 1) Some detailed information about experimental apparatus should be added.
- 2) Comparisons between measured data and simulation results show very good agreement. However, only one figure is provided. In order to confirm the result, more comparison in different should be given.

Reviewer #3:

Remarks to the Author:

The paper claims that thermal diffusion, described by the macroscopic heat equation, is reciprocal under temporal modulation of the coefficients of the equation. The demonstration is both analytical, for a one-dimensional heat channel, and an experimental verification with a particular system implementing one of the two types of modulation that was identified theoretically. The discussion attributes reciprocity to the combination of a material derivative in the heat equation (convection term) and the conservation of mass.

I was globally convinced by the analysis, that relies on a purely classical macroscopic approach of diffusion equations (and not on thermodynamics). The fact that a material derivative must be used if there is a transport of mass (a temporal modulation of the mass density) did not appear as a surprise to this reviewer; indeed it seems a well known fact in heat transport theory. As such, the paper would correct a wrong claim in Ref. [42] (lines 75 and 94), that used the heat equation without the convection correction and concluded that non-reciprocity is expected. This clarification is important but is not clearly identified in the conclusion [if true, it must be stated clearly; if the authors are not completely sure of the conclusion they should at least open the debate because one experiment demonstrating reciprocity does not demonstrate that all experiments would be reciprocal].

It is difficult to state on the potential interest of the paper on a field wider than heat transport and time-modulated systems. It may engender a series of other theories or experimental trials, or just close the topic.

I have the further technical comments for the authors to consider in a revision.

1. Do you agree that equation (2) is a correction of (1) replacing a partial derivative by a material derivative (a total derivative with respect to time)? Using this term could help place the contribution in a better context.
2. I miss the derivation of the in-line equations at lines 128 and 186. They are actually very

important to the conclusions on reciprocity, because their particular form implies the final result. 3. Bloch's theorem is invoked on lines 153-156, but for an ordinary differential equation shouldn't the reference be to Floquet's theorem instead? This is a formal comment since it clearly won't change the derivation.

Reviewer #4:

Remarks to the Author:

In this paper, the authors discuss the possibility of reciprocity of thermal diffusion in time-modulated systems. If the time-modulation shows the reciprocity, it can be used for thermal devices, such as thermal diode, hence it is crucial to figure out this problem.

The authors show the time-dependent Fick's law in (1). In the previous paper the reference [42] Phys. Rev. Lett. 120, 125501 (2018), the equation can reveal the reciprocity. However, the reference [42] assumes arbitrary change on the density field on the mass, not caring on the mass conservation law. In this paper, the authors investigated the new equation (2) employing the mass conservation. As a result, within practically allowed modulation on the parameters, the authors found that no reciprocity can occur. In addition to analytical approach, they demonstrate experimental verification on this.

While the answer is negative on the reciprocity, the finding is honest and important for the field of the thermal engineering. On this aspect, I appreciate very much on the result. One point that I worry is on theoretical rigor. While they show nonreciprocity at 'practical level'. What is a correct result on the reciprocity, if one can change the parameters arbitrary in the equation (2) ? We could not understand if their result is rigorous or approximation. Practical parameters we usually mean can change depending on the experimental development.

We thank all the reviewers for the comments and valuable advice. We have made a lot of changes in the revised manuscript and Supplementary Information, highlighted in blue. We also have added experiments under different parameters and further analysis to confirm the result. Corresponding replies to each reviewer are as follows.

Reply to Reviewer #1:

Authors present a work on “Reciprocity of thermal diffusion in time-modulated systems”. From the abstract, they badly formulate the motivation by saying “Quite surprisingly, here we show that, from a practical point of view, time modulation cannot generally be used to break reciprocity for thermal diffusion.” Practicability does not contradict the possibility or not!

Our reply: We apologize that we did not formulate the motivation properly and that “from a practical point of view” might have caused confusion. We should use a more appropriate expression to interpret the motivation better. The purpose of this paper is to illustrate the deep physical reasons on why nonreciprocal thermal diffusion in time-modulated systems may not occur, in contrast to the widely held captivation otherwise. Admittedly, time modulation of parameters is possible to generate nonreciprocity at the level of a mathematical equation (when the material density and conductivity are set as space- and time-dependent functions). But it is very challenging, if not completely impossible, to find a corresponding physical system to realize such assumptions. We shall mention that the mass motion must be involved in the modulated process and will inevitably change the governing equation, making it a convection-diffusion process that deviates from previous theories. This is the main argument of this article, and we have revised the manuscript accordingly.

Define the reciprocity mathematically in diffusion. It is not so clear compare to EM or acoustics. There is no wavevector and thus you need more to define the problem you solve and in particular the BC (even the orthogonal once to the direction of heat flux).

Our reply: We appreciate the reviewer for mentioning the mathematical definition of reciprocity in diffusion. Due to the lack of directionality of diffusion at steady state, no quantity like scattering coefficient in EM and acoustics can clearly define reciprocity/nonreciprocity in diffusion. In order to solve this problem, we defined the global reciprocity of heat diffusion based on the frequency-domain Green’s function in a recent work titled “Diffusive nonreciprocity and thermal diode” (*Phys. Rev. B.* **103**, 014307). In that work, we established the definition of diffusive reciprocity for a two-port system with different boundary conditions and proved the equivalence between steady-state global nonreciprocity and thermal diode effect. In this work, we apply fixed temperature boundary conditions $T(0,t) = T_{\text{cold}}$ and $T(L,t) = T_{\text{hot}}$ (backward) or $T(0,t) = T_{\text{hot}}$ and $T(L,t) = T_{\text{cold}}$ (forward) at two ends of the system, in order to analyze the reciprocity by comparing forward and backward heat fluxes, and thermal reciprocity requires: $\langle q_{b,1} \rangle = -\langle q_{f,2} \rangle$ and $\langle q_{b,2} \rangle = -\langle q_{f,1} \rangle$. We have added this part and the mathematical definition of thermal reciprocity (see Eq.(4)) in the revised manuscript. We also give the results of the heat flux through (into and out of) the system in Supplementary Figs. 2, 3 and 5 to further illustrate the thermal reciprocity.

What about the parity of the system?

Our reply: Thanks for reminding us to consider the parity of the system. We find that even if the material parameters are not parity-symmetric, the heat transfer system is still reciprocal. The proof is as follows:

Given the density $\rho(x,t) = \rho_a(x) + \rho_b(\chi)$ and thermal conductivity $\kappa(x,t) = \kappa_a(x) + \kappa_b(\chi)$ of the material, where ρ_b and κ_b are periodic functions of $\chi = x - v_0 t$, at least one of ρ_a and κ_a would be dependent on x . Then the system is asymmetric in the spatial distribution of material parameters. According to the 1D continuity equation $\partial\rho/\partial t + \partial(\rho v)/\partial x = 0$, the mass flux ρv satisfies $\rho v = (\rho_b - \rho_{b0})v_0 + C$, where ρ_{b0} is the average of ρ_b and C is a constant (no accumulate mass flux requires $C = 0$). From Eq. (2) in the main text, the 1D heat transfer then obeys

$$\rho c \frac{\partial T}{\partial t} + (\rho_b - \rho_{b0}) c v_0 \frac{\partial T}{\partial x} = \frac{\partial}{\partial x} \left(\kappa \frac{\partial T}{\partial x} \right) \quad (\text{A1})$$

We apply fixed temperature boundary conditions $T(0,t) = T_{\text{cold}}$ and $T(L,t) = T_{\text{hot}}$ (backward) or $T(0,t) = T_{\text{hot}}$ and $T(L,t) = T_{\text{cold}}$ (forward) at the two ends, respectively. Assuming that $T_b(x,t)$ is the solution for the backward case with the boundary conditions: $T_b(0,t) = T_{\text{cold}}$, $T_b(L,t) = T_{\text{hot}}$, while $T_f(x,t)$ is the solution for the forward case with the boundary conditions: $T_f(0,t) = T_{\text{hot}}$, $T_f(L,t) = T_{\text{cold}}$. Given the initial conditions, both solutions should be unique. Their summation $T_s(x,t) = T_f(x,t) + T_b(x,t)$ also satisfies Eq. (A1) with boundary conditions $T_s(0,t) = T_s(L,t) = T_{\text{hot}} + T_{\text{cold}}$. It is easy to check that $T_s(x,t) = T_{\text{hot}} + T_{\text{cold}}$ is a solution, and must be the unique solution thanks to the uniqueness of $T_f(x,t)$ and $T_b(x,t)$. The heat flux $q(x,t)$ is the sum of conductive and convective heat flux: $q(x,t) = -\kappa \partial T / \partial x + \rho c v (T - T_{\text{ref}})$, where T_{ref} is the reference temperature (We set $T_{\text{ref}} = T_{\text{cold}}$). Then the forward and backward heat fluxes $q_f(x,t)$ and $q_b(x,t)$ then satisfy

$$q_f(x,t) + q_b(x,t) = -\kappa \frac{\partial T_s}{\partial x} + (\rho_b - \rho_{b0}) v_0 c (T_s - 2T_{\text{ref}}) = (\rho_b - \rho_{b0}) v_0 c (T_{\text{hot}} - T_{\text{cold}}) \quad (\text{A2})$$

Averaging over time gives $\langle q_f(x) \rangle + \langle q_b(x) \rangle = 0$. Considering that the average heat fluxes in and out of the system should balance, we have $\langle q_f(0) \rangle = \langle q_f(L) \rangle = -\langle q_b(0) \rangle = -\langle q_b(L) \rangle$, which meets the condition for a symmetric heat transfer and indicates 1D thermal reciprocity (The proof for 2D/3D case can be obtained in the same way). To further verify the conclusion, we give an example of a parity-asymmetric system in the Supplementary Fig. 6. Numerical results of the heat transport show that this system is thermally reciprocal.

I find the paper interesting but very badly presented and I get almost not convinced by the experimental part that is not well explained. Must be reformulated.

Our reply: We apologize that the experimental part were not well presented and explained in the previous version. And for that, we have made two improvements. First, in order to reduce the influence of ambient air convection, we set the experimental system in a vacuum chamber, as shown in Figure R1. With this improved equipment, the experimental results are much better than before. Second, we have provided further experimental data and analysis in Fig. 4d and e. The average of temperature difference (ΔT_1 and ΔT_2) over time shows a good symmetric relationship: $\langle \Delta T_{b,1} \rangle \approx -\langle \Delta T_{f,2} \rangle$ and $\langle \Delta T_{b,2} \rangle \approx -\langle \Delta T_{f,1} \rangle$, which can be used to reflect the reciprocity of average heat flow through the intervals. In order to further show the reciprocity, numerical results of the heat flow Q in and out of the system used in the experiment are given in

Supplementary Fig. 5. It is easy to check that $\langle Q_{b,1} \rangle = -\langle Q_{f,2} \rangle$ and $\langle Q_{b,2} \rangle = -\langle Q_{f,1} \rangle$, then validating the thermal reciprocity at the level of definition of (Eq.(4)).

In addition, we have made a thorough revision in the revised manuscript and Supplementary Information, verifying the thermal reciprocity sufficiently, including:

- Theoretical proof: We have added more explanations in the main text. For the first type of density modulation, "...we have $\langle q_f(0) \rangle = \langle q_f(L) \rangle = -\langle q_b(0) \rangle = -\langle q_b(L) \rangle$, which meets the condition for a symmetric heat transfer as in Eq. (4), and indicates thermal reciprocity". For the second type, "...we can prove the thermal reciprocity by analyzing the time-averaged heat fluxes along x direction in forward and backward regimes, which also gives $\langle q_f \rangle + \langle q_b \rangle = 0$ and satisfy Eq. (4)."
- Analytical solutions: In the Supplementary Note 2-4, we give the detailed process to obtain the analytical solutions of the governing equations under time modulation. We also add the formula of heat flux under 1D density modulation (see Eqs.(S33) and (S37)), based on which the time-averaged heat flux are calculated: "For $\mu = 1/d$, the analytical value of time-averaged heat flux density is $\langle q_b \rangle = -2.54 \times 10^4 \text{ W m}^{-2}$ (backward) and $\langle q_f \rangle = 2.54 \times 10^4 \text{ W m}^{-2}$ (forward), while for $\mu = 4/d$, $\langle q_b \rangle = -4.26 \times 10^4 \text{ W m}^{-2}$ (backward) and $\langle q_f \rangle = 4.26 \times 10^4 \text{ W m}^{-2}$ (forward). It is easy to find that $\langle q_b \rangle = -\langle q_f \rangle$, satisfying Eq. (4) in the main text and demonstrating reciprocity in heat transfer."
- Simulated results: We have added simulated results of backward and forward heat flux, see Supplementary Fig. 2, 3, 5 and 6. For example, we give the heat flux distributions of the system for 1D case (Supplementary Fig. 2) and total heat flow at two ports for 3D case (Supplementary Figs.3 and 5). All these simulations show that physical systems preserve thermal reciprocity.
- Experimental results: We have added improved experimental results and discussions, showing symmetry in backward and forward temperature distributions as well as average heat flows, as mentioned above.

We hope to have presented our work better.

Figure. R1 Structural design of the vacuum chamber. **1.** The experimental system is set inside the vacuum chamber ($430 \times 285 \times 260 \text{ mm}^3$), and the lining of the vacuum chamber is coated to reduce the reflectivity. **2.** Two observation windows are fabricated to measure the temperature profiles of the whole system. The size of arms is $270 \times 175 \times 240 \text{ mm}^3$ and the diameter of the circular germanium glass windows is 82 mm. **3.** The rotator driving the system is set outside the vacuum chamber for better heat dissipation. **4.** Supporting platform. **5.** Molecular pump LF-110 and mechanical pump BSV-16 are employed to reduce the density of air in the vacuum chamber as low as 10^{-3} Pa .

Reply to Reviewer #2:

This work suggests that thermal reciprocity has more fundamental resilience than other transport mechanisms. The findings will be important for the thermal system design and other dissipative wave propagation systems. I suggest to publish this paper in Nature Communication after minor revision with following comments.

Our reply: Thanks very much for the positive comments.

1) Some detailed information about experimental apparatus should be added.

Our reply: Thanks for this valuable advice, we have added detailed information about the experimental apparatus in the revised manuscript. In addition, we have improved the experimental equipment using a vacuum chamber (Figure. R1) to reduce the influence of ambient air convection.

2) Comparisons between measured data and simulation results show very good agreement. However, only one figure is provided. In order to confirm the result, more comparison in different should be given.

Our reply: We appreciate the reviewer's suggestion. On the basis of the improved experimental equipment, we have provided further experimental data and analysis to demonstrate the thermal reciprocity. We have added measured temperature difference changing with time in Fig. 4d and e. The symmetry in time-averaged ΔT_1 and ΔT_2 can be seen, indicating the reciprocity of heat flow. For the measured data and simulation results show good agreement, we can estimate the heat flow in and out of the system by means of simulation (see Supplementary Fig. 5), which also confirm the reciprocity of heat flow: $\langle Q_{b,1} \rangle = -\langle Q_{f,2} \rangle$ and $\langle Q_{b,2} \rangle = -\langle Q_{f,1} \rangle$. The angular velocity is selected as $0.075\pi \text{ rad s}^{-1}$ (Fig. 4d and Supplementary Fig. 5a) and $0.14\pi \text{ rad s}^{-1}$ (Fig. 4e and Supplementary Fig. 5b), as two representative cases of the general situation where the modulation parameters (e.g. rotating angular velocity) are changed arbitrarily.

Reply to Reviewer #3:

The paper claims that thermal diffusion, described by the macroscopic heat equation, is reciprocal under temporal modulation of the coefficients of the equation. The demonstration is both analytical, for a one-dimensional heat channel, and an experimental verification with a particular system implementing one of the two types of modulation that was identified theoretically. The discussion attributes reciprocity to the combination of a material derivative in the heat equation (convection term) and the conservation of mass.

I was globally convinced by the analysis, that relies on a purely classical macroscopic approach of diffusion equations (and not on thermodynamics). The fact that a material derivative must be used if there is a transport of mass (a temporal modulation of the mass density) did not appear as a surprise to this reviewer; indeed it seems a well known fact in heat transport theory. As such, the paper would correct a wrong claim in Ref. [42] (lines 75 and 94), that used the heat equation without the convection correction and concluded that non-reciprocity is expected. This clarification is important but is not clearly identified in the conclusion [if true, it must be stated clearly; if the authors are not completely sure of the conclusion they should at least open the debate because one experiment demonstrating reciprocity does not demonstrate that all experiments would be reciprocal].

Our reply: We appreciate the reviewer's precise and positive comments, and apologize for the delay in responding for we have spent long time on improving the experimental equipment. Although it is well known that a material derivative must be used if there is a transport of mass, this fact might be overlooked when both heat transfer and time modulation are involved, especially when the researcher's main research field is not in heat transfer. In addition, the specific effects of time modulation on heat transfer are not clear by now, and it is studied in this paper, which is necessary, timely and important.

We agree with reviewer's statement and apologize for not making the conclusions clear. We confirm the conclusion in the revised manuscript "In this paper, we have shown that physical systems preserve thermal reciprocity under time modulation as a result of a convection correction introduced by density modulation...For heat transport, however, it is a well-known fact that a material derivative must be used if there is a transport of mass. As such, a convection correction should be added in the heat transfer equation, resulting in the preservation of thermal reciprocity..." We believe that there are no more debates about this conclusion because our theory is rigorous. Although only representative experiments are presented, theoretical proof and numerical results are provided to confirm our conclusion. While there is no debate, we propose other possible ways to break the reciprocity, which are beyond our assumptions and answered in the next question.

It is difficult to state on the potential interest of the paper on a field wider than heat transport and time-modulated systems. It may engender a series of other theories or experimental trials, or just close the topic.

Our reply: Thanks for raising the question. Heat transfer and time modulation are two research fields broad and stable enough, and this paper may have contributions to both of them. Our work

is of fundamental interest to those concerned with the symmetry of heat transfer. We find a unique mechanism that heat transfer preserves the reciprocity under time modulation, which provides a reference for researchers trying to break reciprocity and build thermal diodes thereafter. For time modulation, the influence of practical modulation behavior has been ignored. This work may serve as a first “strong voice” that the physics of modulation (aka, underlying mechanisms, restrictions, and validity of theories) should be deeply checked and considered during the design of time-modulated systems (which is a very hot topics nowadays), not only for the field of heat transport, but also applicable to other diffusive and wave phenomena. Therefore, this work may attract interest from both of the two fields.

Although we present a somehow negative conclusion on thermal nonreciprocity under time modulation, we do not aim to close the topic. Firstly, we find that physical systems preserve thermal reciprocity as a result of a convection correction introduced by density modulation, which is an interesting and somewhat surprising conclusion, considering the continuous success of nonreciprocal effect by time modulation in electromagnetics, photonics, and acoustics and charge diffusion. Secondly, there are schemes that can break thermal reciprocity beyond our assumptions. In the revised manuscript, we add discussions on the promising ways to generate thermal nonreciprocity or design thermal diodes. One way is to use materials undergoing phase transitions in the presence of an electric or magnetic field to enable, in principle, a change in heat capacity without mass motion. In addition, though we claim that thermal reciprocity is preserved as the system boundaries are maintained at two constant temperatures, we find that thermal wave nonreciprocity may occur when the boundaries are set as periodically oscillating heat sources. Another trivial way is to introduce directional energy bias while modulating the parameters, such as choosing an external force f , the work done by which is different in forward and backward directions. Even if there are challenges to implementing these approaches, they may provide ideas to realize nonreciprocal thermal management.

I have the further technical comments for the authors to consider in a revision.

1. Do you agree that equation (2) is a correction of (1) replacing a partial derivative by a material derivative (a total derivative with respect to time)? Using this term could help place the contribution in a better context.

Our reply: Agree. Thank the reviewer for this valuable advice. We have used this term in the revised manuscript and Supplementary Note 1.

2. I miss the derivation of the in-line equations at lines 128 and 186. They are actually very important to the conclusions on reciprocity, because their particular form implies the final result.

Our reply: The detailed derivation of the in-line equation at line 128 is recorded in Supplementary Note 2 (Eqs. (S14)-(S16)). The derivation for the equation at line 186 can be obtained in the same way. Thanks for reminding us and we have added corresponding annotations in the revised manuscript.

3. Bloch's theorem is invoked on lines 153-156, but for an ordinary differential equation shouldn't the reference be to Floquet's theorem instead? This is a formal comment since it clearly won't change the derivation.

Our reply: The solutions of the type exhibited on line 155 has been known to mathematicians as Floquet's theorem, which deals with the solution of 1D partial differential equations with periodic coefficients. Bloch generalized Floquet's results to 3D systems and obtained the description of the wave function when electrons in periodic lattices are being discussed, so the equation on line 155 is usually described as the Bloch theorem and the functions themselves are called Bloch functions. It is often stated that Bloch's theorem and Floquet's theorem are equivalent. Thanks for reminding us, and we refer it as Floquet-Bloch theorem instead in the revised manuscript.

Reply to Reviewer #4:

In this paper, the authors discuss the possibility of reciprocity of thermal diffusion in time-modulated systems. If the time-modulation shows the reciprocity, it can be used for thermal devices, such as thermal diode, hence it is crucial to figure out this problem.

The authors show the time-dependent Fick's law in (1). In the previous paper the reference [42] Phys. Rev. Lett. 120, 125501 (2018), the equation can reveal the reciprocity. However, the reference [42] assumes arbitrary change on the density field on the mass, not caring on the mass conservation law. In this paper, the authors investigated the new equation (2) employing the mass conservation. As a result, within practically allowed modulation on the parameters, the authors found that no reciprocity can occur. In addition to analytical approach, they demonstrate experimental verification on this.

While the answer is negative on the reciprocity, the finding is honest and important for the field of the thermal engineering. On this aspect, I appreciate very much on the result.

Our reply: Thanks very much for the positive comments. We also apologize for the delay in responding, because the manufacture of the improved experimental equipment has taken a lot of time.

One point that I worry is on theoretical rigor. While they show nonreciprocity at 'practical level'. What is a correct result on the reciprocity, if one can change the parameters arbitrary in the equation (2) ? We could not understand if their result is rigorous or approximation. Practical parameters we usually mean can change depending on the experimental development.

Our reply: We apologize that the wording “practical” may have caused misunderstanding. What we try to express is the physical infeasibility of nonreciprocal thermal diffusion in time-modulated systems. This paper rigorously demonstrates that the thermal reciprocity cannot be broken by spatiotemporal modulation, taking into account the practice of density modulation. The conclusion is theoretically rigorous and universal for arbitrary parameters, for the symmetric relationship between the forward and backward heat flux $\langle q_f \rangle = -\langle q_b \rangle$ (proved in the main text) always exists, regardless of how the material parameters are modulated. The proof holds for arbitrary parameters. Though only several examples (simulation and experiment) are given in the paper, indeed we have attempted various kinds of modulation of parameters in the simulations and experiments to demonstrate the absence of nonreciprocity in all circumstances. The reciprocal result exhibited in this work is not a special case, but a representative of the general situation where the modulation parameters are changed arbitrarily. Thanks for reminding us and we have add explanations and evidences in the revised manuscript to verify the thermal reciprocity sufficiently (including theoretical proof, analytical solutions, simulated results and experimental results).

Reviewers' Comments:

Reviewer #1:

Remarks to the Author:

Authors have responded to my comments and the paper can be accepted in the current form.

Reviewer #3:

Remarks to the Author:

I was globally convinced by the answers to the comments and the revisions that were made to the manuscript. The supplementary information text mostly has been significantly expanded to include more details regarding theory and experiment. This supplementary text should be carefully checked against typos since it contains many English mistakes. The new equations are not so easy to follow, but the theoretical derivations appear to be confirmed by numerical simulations and experiments, so I was convinced that they are globally correct.

Overall, I wish to congratulate the authors for the efforts spent on the revision and I have no further technical questions.

Reviewer #4:

Remarks to the Author:

The authors replied to my comments. My request was to formulate correctly and clarify if the reciprocity results are rigorous or not.

They correctly replied to these requests, and so I would accept the draft for publication with the present form.

Response to the Reviewers' Comments on the manuscript NCOMMS-20-50861A entitled "Reciprocity of thermal diffusion in time-modulated systems" submitted to Nature Communications

We thank all the reviewers again for their careful review and valuable advice. The comments are useful and allow us to further refine the manuscript. Corresponding replies to each reviewer are as follows:

Reviewer #1 (Remarks to the Author):

Authors have responded to my comments and the paper can be accepted in the current form.

Our reply: We are thankful to the reviewer for the recommendation for publication.

Reviewer #3 (Remarks to the Author):

I was globally convinced by the answers to the comments and the revisions that were made to the manuscript. The supplementary information text mostly has been significantly expanded to include more details regarding theory and experiment. This supplementary text should be carefully checked against typos since it contains many English mistakes. The new equations are not so easy to follow, but the theoretical derivations appear to be confirmed by numerical simulations and experiments, so I was convinced that are globally correct.

Overall, I wish to congratulate the authors for the efforts spent on the revision and I have no further technical questions.

Our reply: Thanks very much for the positive comments and the valuable advice. We have revised the supplementary text carefully to avoid English mistakes.

Reviewer #4 (Remarks to the Author):

The authors replied to my comments. My request was to formulate correctly and clarify if the reciprocity results are rigorous or not.

They correctly replied to these requests, and so I would accept the draft for publication with the present form.

Our reply: We thank the reviewer for reviewing carefully and recommending publication.